

# Factors related to overweight and obese populations maintaining metabolic health

Yi-Hsuan Lin[1,2], Hsiao-Ting Chang[2,3], Yen-Han Tseng[2,4], Harn-Shen Chen[2,5], Shu-Chiung Chiang[6], Tzeng-Ji Chen[2,3,6] and Shinn-Jang Hwang[2,3]

[1] Department of Family Medicine, Cheng Hsin General Hospital, Taipei, Taiwan
[2] Faculty of Medicine, School of Medicine, National Yang Ming Chiao Tung University, Taipei, Taiwan
[3] Department of Family Medicine, Taipei Veterans General Hospital, Taipei, Taiwan
[4] Department of Chest Medicine, Taipei Veterans General Hospital, Taipei, Taiwan
[5] Division of Endocrinology and Metabolism, Department of Internal Medicine, Taipei Veterans General Hospital, Taipei, Taiwan
[6] Institute of Hospital and Health Care Administration, School of Medicine, National Yang Ming Chiao Tung University, Taipei, Taiwan

## ABSTRACT

**Background**. For people who are overweight or obese, maintaining a metabolically healthy status can decrease the risks of developing cardiovascular diseases and Type 2 diabetes. Despite this, only a limited amount of research has discussed the metabolically healthy overweight and obesity (MHOO) population in Asia and the factors associated with them maintaining their metabolic health.

**Methods**. This study enrolled 195 MHOO participants from communities in northern Taiwan during 2009–2010 (baseline). Of the 195 participants, 89 completed the follow-up assessment after a median follow-up time of nine years. Body type was determined by body mass index (BMI, $kg/m^2$). We defined overweight as a BMI $\geq$ 24 $kg/m^2$ and <27 $kg/m^2$ and defined obese as a BMI $\geq$ 27 $kg/m^2$. Metabolic health was defined as the absence of cardiometabolic diseases and the presence of $\leq$1 of the cardiometabolic risk factors, namely hypertension, hyperglycemia, hypertriglyceridemia, and low serum high-density lipoprotein cholesterol. Metabolic health, BMI, and other covariates were evaluated at both baseline and follow-up. Generalized estimating equations (GEE) models were used to analyze the factors associated with maintenance of metabolic health during the follow-up period.

**Results**. At baseline, the mean age of the study participants was 47.4 (SD 5.3) years and 46 (51.7%) of the participants were women. There were 51 (57.3%) individuals who maintained their metabolic health status at the time of the nine-year follow-up. The detrimental factors pertaining to metabolic health included older age, longer duration until follow-up, BMI $\geq$ 27 $kg/m^2$, and increase in waist circumference. No significant relationships were observed between sociodemographic factors and lifestyle factors, such as sex, level of education, cigarette smoking, alcohol consumption, and physical activity, and sustained metabolic health among MHOO individuals.

**Conclusions**. To maintain metabolic health and prevent negative changes in health status, control of bodyweight and waist circumference should remain a priority for MHOO individuals even when there are no metabolic disorders present.

Corresponding author
Hsiao-Ting Chang,
htchang2@vghtpe.gov.tw

## INTRODUCTION

Being obese affects major dimensions of health, including the physiological, psychological, and sociological. In addition to experiencing greater risks of cardiovascular diseases, Type 2 diabetes, and mortality (*Kinlen, Cody & O'Shea, 2017*), obese people were more vulnerable to depression (*Dixon, Dixon & O'Brien, 2003*) and employment discrimination (*Flint et al., 2016*). Additionally, obesity results in substantial health care burdens worldwide for both inpatient and outpatient costs (*Tremmel et al., 2017*).

A subgroup of obese people not experiencing metabolic disorders, referred to as metabolically healthy obesity (MHO), has attracted extensive attention in recent years. Obesity typically has been measured by body mass index (BMI), waist circumference, and total body fat percentage (*Rey-López et al., 2014*). The criteria for metabolic health has typically included the absence of hypertension, hyperlipidemia, hyperglycemia, abdominal obesity, and insulin resistance, though some studies have added glycated hemoglobin, uric acid, C-reactive protein, and white blood cell count in addition to these metabolic health indicators (*Rey-López et al., 2014*). Researchers using different definitions of MHO has resulted in a lack of standardization in studies of MHO. As such, a universal definition of MHO is needed to facilitate comparability between studies.

Much research has discussed the health impact of MHO. Some cohort studies have reported individuals with MHO had higher risks of cardiovascular diseases and Type 2 diabetes compared with people of metabolically healthy normal weight (*Kim et al., 2012*; *Yeh et al., 2021*); however, it is noted these studies only evaluated metabolic health and obesity at baseline without any follow-up. Other studies have considered change of metabolic health and BMI over time and have yielded no significant associations between sustained MHO status and incident diabetes and cardiovascular events (*Appleton et al., 2013*; *Gao et al., 2020*). For people experiencing obesity, maintaining a good metabolic health status helps to avoid poorer health outcomes. A question that arises from this: what are the characteristics of those MHO individuals who maintain a positive metabolic health status over time?

Studies from the United States, United Kingdom, and Australia have investigated the factors associated with changes in metabolic health status among MHO individuals (*Appleton et al., 2013*; *Achilike et al., 2015*; *Moussa et al., 2019*). A younger age, non-smoker status, better insulin sensitivity, higher social-economic status, lower BMI, and smaller waist circumference have been reported to be related to a sustained metabolically healthy status for MHO. Studies of Asian populations have been rare for this topic. *Hwang et al. (2015)* analyzed a cohort of second and third generation Japanese Americans and reported some characteristics (male, less visceral abdominal fat, higher level of high-density lipoprotein cholesterol (HDL-C), and lower level of fasting plasma insulin) were related to sustained metabolically healthy status for obese people. However, data originating in Asia is lacking,

and the factors associated with maintenance of metabolic health among Asian people with MHO are unknown.

This study aimed to explore the characteristics related to the maintenance of metabolic health for individuals with metabolically healthy overweight and obesity (MHOO) among the Taiwanese population. We used a community-based, prospective cohort setting. By exploring the associated factors related to the maintenance of metabolic health for Taiwanese people, we hoped to add to the literature base for a different ethnicity to gain a deeper understanding of MHO and to shed light on health promotion for obese people.

## MATERIALS & METHODS

### Study design, participants, and sample size

This study was a prospective cohort study. The protocol of this study was approved by the Research Ethics Committee of Taipei Veteran General Hospital (Protocol Code: 2017-01-009BCF, 97-12-06A). The execution of the study followed the rules of the Research Ethics Committee. During 2009–2010 (baseline), we enrolled the study population from residents living in the Shipai area (Shilin and Beitou district) of Taipei, Taiwan. The inclusion criteria were (1) residents living in the Shipai area for more than six months and (2) residents aged 35–55 years old. We applied for permission from the Taipei City Bureau of Civil Affairs for the assessment of household registration information. Subjects were randomly recruited from neighborhoods and an invitation to participate in this research was sent to the eligible residents. Residents who agreed to participate in this study signed an informed consent form before recruitment (*Chiang et al., 2018*). At baseline, there were 906 participants enrolled in this study. Among them, 711 were excluded who did not satisfy the definition of MHOO in this study. Only 195 participants with MHOO at baseline were preserved for analyses. After a median follow-up period of 9 years (range: 8–10 years), 89 participants completed the follow-up between 2017 and 2018 (Fig. 1). We observed the status of metabolic health after nine years and the associated factors.

### Measurements of overweight, obesity and metabolic health

The measurements of overweight, obesity, and metabolic health were evaluated at baseline and the nine-year follow-up. Participants' body types were determined by BMI ($kg/m^2$). In accordance with the Health Promotion Administration, Ministry of Health and Welfare in Taiwan (*Health Promotion Administration, Ministry of Health and Welfare in Taiwan, 2018*), overweight was defined as $24 \text{ kg/m}^2 \leq \text{BMI} < 27 \text{ kg/m}^2$ and obesity was defined as $\text{BMI} \geq 27 \text{ kg/m}^2$.

The definition of metabolic health was based on the consensus definition for metabolic syndrome (*Alberti et al., 2009*). We adopted more stringent criteria (*Smith, Mittendorfer & Klein, 2019*), compromised of (A) an absence of known cardiometabolic diseases, including hypertension, hyperlipidemia, Type 2 diabetes, coronary artery diseases, cerebral vascular diseases, and peripheral vascular diseases and (B) the presence of ≤1 of the following cardiometabolic profiles: (1) hypertension: blood pressure ≥130/85 mmHg or on drug treatment, (2) hyperglycemia: serum fasting glucose ≥ 100 mg/dl or receiving drug treatment, (3) hypertriglyceridemia: serum triglyceride ≥ 150 mg/dl or receiving drug

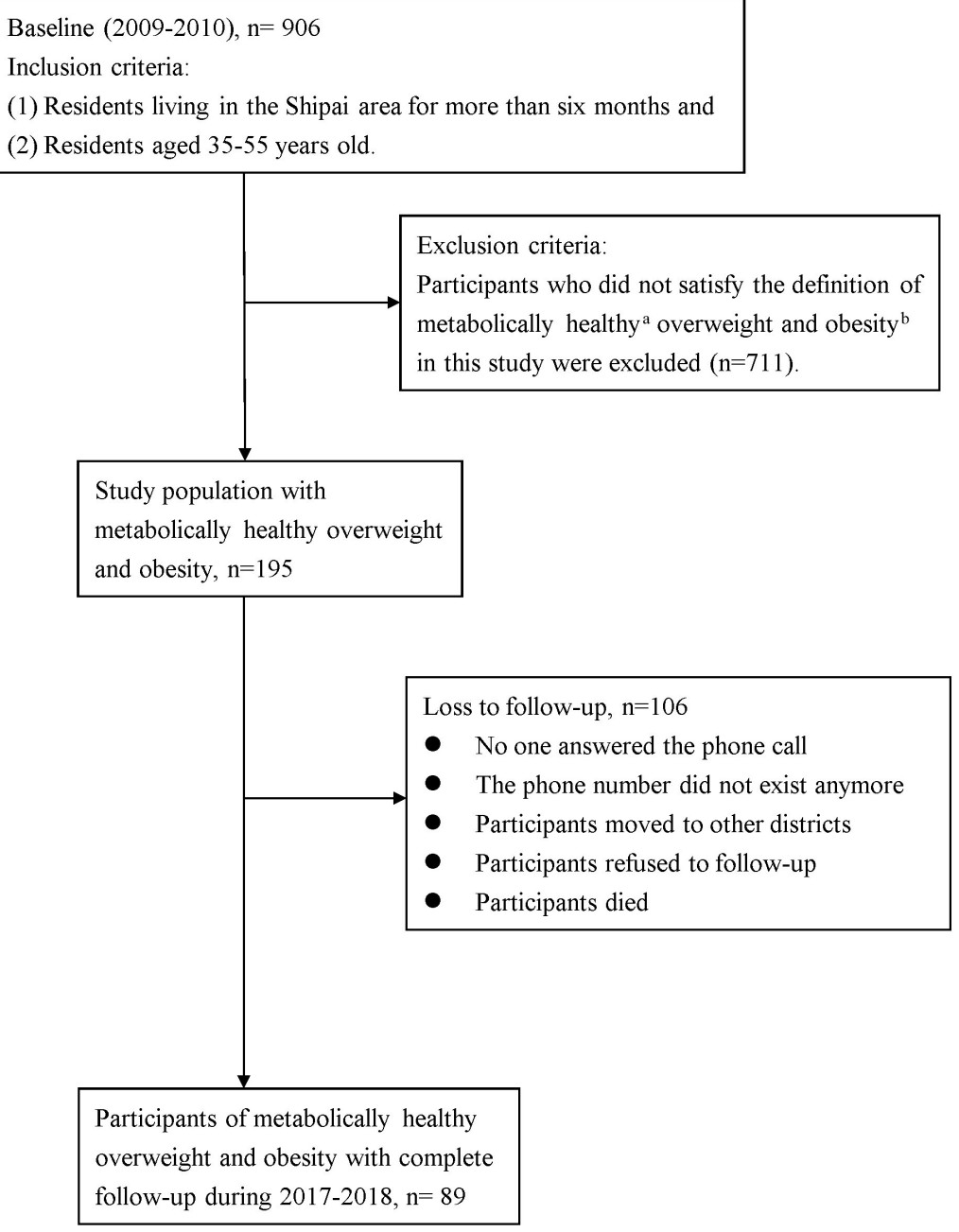

**Figure 1** **Flow chart of study design.** (A) Metabolic health was defined as (1) the absence of cardiometabolic diseases and (2) the presence of ≤1 of the cardiometabolic risk factors, including hypertension, hyperglycemia, hypertriglyceridemia, and low serum high-density lipoprotein cholesterol. (B) Overweight was defined as 24 kg/m² ≤ BMI < 27 kg/m² and obesity was defined as BMI ≥ 27 kg/m².

treatment, and (4) low serum HDL-C: HDL-C < 40 mg/dL for men and <50 mg/dL for women or receiving drug treatment.

Blood pressure while in a seated position was checked twice to obtain an average blood pressure. Waist circumference was measured at the midpoint between the lowest rib and iliac crest at the end of expiration. Blood tests for each participant were performed when they had been in a fasting state for more than eight hours. Blood chemistry analyses including serum glucose, triglyceride, total cholesterol, low-density lipoprotein cholesterol, and HDL-C were conducted at the central laboratory of Taipei Veteran General Hospital.

## Other variables

Participants reported their basic demographic data at baseline, including age, sex, marital status, medical history, and level of education. Other lifestyle factors, including cigarette smoking, alcohol consumption, and physical activity, were evaluated via questionnaires. The information of metabolic health, BMI, waist circumference, cigarette smoking, alcohol consumption, and physical activity were evaluated at both baseline and follow-up assessment. Cigarette smoking was categorized as non-smoker, current smoker, or ex-smoker. We evaluated alcohol consumption by asking participants whether they regularly drink alcohol. Levels of physical activity were measured by the International Physical Activity Questionnaire (IPAQ) Short-Form, Taiwan version (*Liou et al., 2008*). Participants reported the duration and frequency they had spent participating in walking, moderate physical activity, and vigorous physical activity over the past one week. The amount of physical activity was converted into metabolic equivalent units (MET-minutes/week), and the physical activity then was quantified as low, moderate, or high-level physical activity (*The International Physical Activity Questionnaire Group, 2010*). The follow-up year was obtained by subtracting the baseline year from the year of follow-up. The follow-up year ranged from 8–10 years, with a median of 9 years.

## Statistical analyses

We compared the characteristics of MHOO individuals who maintained their metabolic health over time with those who had become metabolically unhealthy. For the demographic characteristics, we performed Chi-Square tests and Fisher's exact tests for categorical variables, and $t$-tests for continuous variables. Because the variables were repeated-measured with autocorrelation, we selected generalized estimating equations (GEE) for the main analyses (*Liang & Zeger, 1986*). The GEE models, which are characterized by their robust standard error, estimate unbiased parameters.

We performed two models to examine the factors associated with the maintenance of metabolic health in MHOO individuals. Model 1 adjusted for baseline age, sex, level of education, cigarette smoking, alcohol consumption, physical activity, follow-up years, and groupings of BMI. Model 2 was the same as Model 1 except it adjusted for waist circumference instead of groupings of BMI. In both statistic models, cigarette smoking, alcohol consumption, physical activity, waist circumference, and groupings of BMI were measured at baseline and follow-up. We further performed subgroup analyses stratified by baseline age to test the effect modification of age. The cut-point was set at 48 years

for the dichotomization of sample size. Because seven tests were performed in this study, we used the Bonferroni method to adjust the type I error rate (*Bland & Altman, 1995*). The original α value was 0.05. After Bonferroni adjustment, the α value was set to 0.007 (0.05/7). Therefore, a two-tailed *p*-value of <0.007 was considered statistically significant. All statistical analyses were performed using SAS version 9.4 (SAS Institute Inc., Cary, NC, USA).

### Sensitivity analyses

Two sensitivity analyses were performed by using a different definition of metabolic health in one and different cut-points for overweight and obesity in the other.

In the first sensitivity analysis, we changed the definition of metabolic health as follows: (1) the absence of cardiometabolic diseases and (2) the presence of ≤ 2 of the metabolic risk factors. In addition to the metabolic risk factors in the main analyses, we further adopted abdominal obesity in the sensitivity analysis. A waist circumference ≥ 90 cm in men and ≥ 80 cm in women for Asian people was regarded as abdominal obesity (*Alberti et al., 2009*).

In the second sensitivity analysis, we adopted the same definition of metabolic health as the main analysis but changed the cut-points for overweight and obesity. We defined overweight as $23 \text{ kg/m}^2 \leq \text{BMI} < 25 \text{ kg/m}^2$ and obesity as $\text{BMI} \geq 25 \text{ kg/m}^2$, in accordance with the Asian population-specific cut-points recommended by the World Health Organization (*World Health Organization. Western Pacific Region, 2000*).

## RESULTS

There were 106 (54.4%) MHOO participants from baseline who were lost to follow-up. The comparisons of the demographic characteristics between participants who completed follow-up and those lost to follow-up are listed in Table S1. The demographic characteristics, health behaviors, and metabolic profiles were similar between these two groups, except that on average participants lost to follow-up had lower blood pressure than those who completed follow-up (mean systolic blood pressure: 119.6 mmHg *vs.* 130.1 mmHg; mean diastolic blood pressure: 79.7 mmHg *vs.* 85.5 mmHg, *p* < 0.001).

Table 1 lists the demographic characteristics of the study population. The mean age of study participants was 47.4 (SD 5.3) years and 46 (51.7%) participants were women. There were 51 (57.3%) MHOO individuals who maintained their metabolic health during the follow-up period. Compared to individuals who became metabolically unhealthy at follow-up, participants with sustained metabolic health were thinner (mean BMI: $26.1 \text{ kg/m}^2$ *vs.* $27.4 \text{ kg/m}^2$, *p* = 0.01; mean waist circumference: 85.0 cm *vs.* 92.3 cm, *p* < 0.001). Participants who sustained metabolic health also had a lower mean systolic blood pressure (122.7 mmHg *vs.* 139.9 mmHg, *p* < 0.001), mean diastolic blood pressure (80.9 mmHg *vs.* 91.8 mmHg, *p* < 0.001), mean fasting glucose (90.4 mg/dl *vs.* 94.6 mg/dl, *p* = 0.02), and mean serum low-density lipoprotein cholesterol (124.3 mg/dl *vs.* 139.3 mg/dl, *p* = 0.02). Otherwise, there were no significant differences between these two groups in age, sex, marital status, level of education, cigarette smoking, alcohol consumption, physical activity, mean total cholesterol, mean triglyceride, and mean HDL-C.

**Table 1  Demographic characteristics of MHOO participants at baseline, stratified by changes in metabolic health at follow-up.**

| | Sustained metabolic health $n = 51$ (57.3%) | Became metabolically unhealthy $n = 38$ (42.7%) | P |
|---|---|---|---|
| Age in years, n (%) | | | |
| 30–39 | 8 (15.7) | 4 (10.5) | |
| 40–49 | 29 (56.9) | 17 (44.7) | 0.24 |
| 50–59 | 14 (27.5) | 17 (44.7) | |
| mean age (SD) | 46.7 (5.2) | 48.3 (5.3) | 0.16 |
| Women, n (%) | 27 (52.9) | 19 (50) | 0.78 |
| Marital status, n (%) | | | |
| Others | 3 (5.9) | 4 (10.5) | 0.45 |
| Married | 48 (94.1) | 34 (89.5) | |
| Education, n (%) | | | |
| Illiterate/elementary/senior/junior high school | 25 (49.1) | 18 (47.4) | 0.88 |
| University and above | 26 (51.0) | 20 (52.6) | |
| Cigarette smoking, n (%) | | | |
| Non-smoker | 41 (80) | 28 (73.7) | |
| Smoker | 6 (12) | 7 (18.4) | 0.75 |
| Ex-smoker | 4 (8) | 3 (7.9) | |
| Alcohol consumption, n (%) | 11 (21.6) | 5 (13.2) | 0.31 |
| Physical activity[a], n (%) | | | |
| Low | 18 (35.3) | 14 (36.8) | |
| Moderate | 21 (41.2) | 18 (47.4) | 0.65 |
| High | 12 (23.5) | 6 (15.8) | |
| BMI (kg/m$^2$) | 26.1 (2.1) | 27.4 (2.7) | 0.01 |
| Waist circumference (cm) | 85.0 (7.8) | 92.3 (10.7) | <0.001 |
| Systolic blood pressure (mmHg) | 122.7 (15.4) | 139.9 (19.4) | <0.001 |
| Diastolic blood pressure (mmHg) | 80.9 (10.6) | 91.8 (12.8) | <0.001 |
| Fasting glucose (mg/dl) | 90.4 (7.5) | 94.6 (9.0) | 0.02 |
| Total cholesterol (mg/dl) | 195.4 (31.7) | 210.3 (40.5) | 0.05 |
| Triglyceride (mg/dl) | 97.2 (51.2) | 118.6 (62.3) | 0.08 |
| HDL-C (mg/dl) | 57.6 (10.9) | 54.3 (10.7) | 0.16 |
| LDL-C (mg/dl) | 124.3 (26.3) | 139.3 (32.8) | 0.02 |

**Notes.**

Abbreviations: MHOO, metabolically healthy overweight/obesity; SD, standard deviation; BMI, body mass index; HDL-C, high-density lipoprotein cholesterol; LDL-C, low-density lipoprotein cholesterol.

[a] Physical activity was evaluated by the International Physical Activity Questionnaire (IPAQ) Short-Form, Taiwan version.

Chi-square tests and Fisher's exact tests were used for categorical variables.

$T$-tests were used for continuous variables.

Table 2 compares the characteristics of study participants at baseline and follow-up (on average, 9 years later). At baseline, nearly one-third of participants with MHOO were obese, and this prevalence increased to 40.5% by the follow-up; meanwhile, 16.9% of participants were no longer overweight and obese. Overall, participants had become more sedentary

**Table 2  Health behaviors and cardiometabolic profiles of MHOO participants at baseline and follow-up ($n = 89$).**

|  | Baseline | Nine-year follow-up | P |
|---|---|---|---|
| Cigarette smoking, n (%) |  |  |  |
| Non-smoker | 69 (77.3) | 74 (83.2) |  |
| Smoker | 13 (14.8) | 8 (9.0) | 0.49 |
| Ex-smoker | 7 (8.0) | 7 (7.9) |  |
| Alcohol consumption, n (%) |  |  |  |
| No | 73 (82.0) | 66 (74.2) | 0.20 |
| Yes | 16 (18.0) | 23 (25.8) |  |
| BMI (kg/m²), n (%) |  |  |  |
| BMI <24 | 0 | 15 (16.9) |  |
| 24 ≤ BMI <27 | 59 (66.3) | 38 (42.7) | <0.001 |
| BMI ≥27 | 30 (33.7) | 36 (40.5) |  |
| mean BMI (SD) | 26.6 (2.5) | 27.0 (3.4) | 0.41 |
| Physical activity[a], n (%) |  |  |  |
| Low | 32 (36.0) | 52 (58.4) |  |
| Moderate | 39 (43.8) | 25 (28.1) | 0.01 |
| High | 18 (20.2) | 12 (13.5) |  |
| Prevalence of cardiometabolic profiles[b], n (%) |  |  |  |
| Abdominal obesity | 65 (73.0) | 54 (60.7) | 0.08 |
| Hypertension | 26 (29.2) | 54 (60.7) | <0.001 |
| Hyperglycemia | 7 (7.9) | 32 (41.0) | <0.001 |
| Low HDL-C | 7 (7.9) | 23 (28.4) | <0.001 |
| Hypertriglyceridemia | 10 (11.2) | 15 (19.2) | 0.15 |

Notes.

Abbreviations: MHOO, metabolically healthy overweight/obesity; BMI, body mass index; SD, standard deviation; HDL-C, high-density lipoprotein cholesterol.

[a] Physical activity was evaluated by the International Physical Activity Questionnaire (IPAQ) Short-Form, Taiwan version.

[b] Abdominal obesity was defined as a waist circumference ≥ 90 cm in men and ≥ 80 cm in women. Hypertension represented blood pressure ≥130/85 mmHg or receiving drug treatment. Hyperglycemia represented fasting glucose ≥ 100 mg/dl or receiving drug treatment. Low HDL-C represented serum HDL-C <40 mg/dL for men and <50 mg/dL for women or receiving drug treatment. Hypertriglyceridemia represented serum triglyceride ≥ 150 mg/dl or receiving drug treatment.

Chi-square tests and Fisher's exact tests were used for categorical variables.

T-tests were used for continuous variables.

(low level of physical activity: 36.0% to 58.4%) after nine years, and the prevalence of hypertension, hyperglycemia, and low HDL-C increased over time (29.2% to 60.7%, 7.9% to 41.0%, and 7.9% to 28.4%, respectively; $p < 0.001$).

Table 3 lists the factors associated with maintenance of metabolic health for MHOO individuals. The factors associated with decreases in metabolic health included a higher age at baseline (model 1: OR = 0.85, 95% CI [0.79–0.92]; model 2: OR = 0.88, 95% CI [0.82–0.95]) and a greater number of years until follow-up (model 1: OR = 0.25, 95% CI [0.12–0.53]; model 2: OR = 0.37, 95% CI [0.18–0.76]). Obesity was a significant risk factor for decreased metabolic health (model 1: OR = 0.12, 95% CI [0.03–0.49], $p = 0.003$). In model 2, the odds of being metabolically healthy decreased approximately 19% with

**Table 3  Factors related to MHOO individuals maintaining metabolic health ($n = 89$).**

| | Model 1 | | Model 2 | |
|---|---|---|---|---|
| | OR (95% CI) | P | OR (95% CI) | P |
| Baseline age | 0.85 (0.79, 0.92) | <0.001 | 0.88 (0.82, 0.95) | <0.001 |
| Women | 0.50 (0.22, 1.13) | 0.09 | 0.42 (0.18, 0.96) | 0.04 |
| Education | | | | |
|   Illiterate/elementary/senior/junior high school | Ref. | | Ref. | |
|   University and above | 0.77 (0.36, 1.65) | 0.50 | 0.73 (0.36, 1.50) | 0.39 |
| Cigarette smoking[a] | | | | |
|   Non-smoker | Ref. | | Ref. | |
|   Smoker | 0.34 (0.11, 1.08) | 0.07 | 0.52 (0.18, 1.48) | 0.22 |
|   Ex-smoker | 0.30 (0.08, 1.16) | 0.08 | 0.51 (0.14, 1.82) | 0.30 |
| Alcohol consumption[a] | 0.74 (0.29, 1.87) | 0.52 | 0.90 (0.36, 2.24) | 0.82 |
| Physical activity[a,b] | | | | |
|   Low | Ref. | | Ref. | |
|   Moderate | 0.81 (0.37, 1.76) | 0.60 | 0.92 (0.44, 1.94) | 0.83 |
|   High | 3.09 (1.07, 8.96) | 0.04 | 2.33 (0.86, 6.28) | 0.10 |
| Follow-up year | 0.25 (0.12, 0.53) | <0.001 | 0.37 (0.18, 0.76) | 0.006 |
| BMI[a] ($kg/m^2$) | | | | |
|   BMI <24 | Ref. | | – | |
|   $24 \leq$ BMI <27 | 1.10 (0.29, 4.10) | 0.89 | – | |
|   BMI $\geq$ 27 | 0.12 (0.03, 0.49) | 0.003 | – | |
| Waist circumference[a] (cm) | – | | 0.91 (0.86, 0.95) | <0.001 |

**Notes.**

Abbreviations: MHOO, metabolically healthy overweight/obesity; OR, odds ratio; CI, confidence interval; BMI, body mass index.

[a]Cigarette smoking, alcohol consumption, physical activity, groupings of BMI, and waist circumference were measured at baseline and follow-up.

[b]Physical activity was evaluated by the International Physical Activity Questionnaire (IPAQ) Short-Form, Taiwan version.

The analyses were performed using generalized estimating equations (GEE) models.

each one cm increase in waist circumference (model 2: OR = 0.91, 95% CI [0.86–0.95], $p < 0.001$).

Table 4 displays the results of the subgroup analyses. The detrimental effects of obesity on maintenance of metabolic health were stronger among MHOO individuals younger than 48 years old than among those more than 48 years old [OR (95% CI) = 0.04 (0.002−0.82) *vs.* 0.08 (0.01−0.63)]; however, these effects did not reach a significant level after the Bonferroni adjustment ($\alpha = 0.007$). The negative effects of years until follow-up also were significant among MHOO individuals younger than 48 years old (OR = 0.04, 95% CI [0.005–0.32], $p = 0.003$).

In the first sensitivity analysis (Table 5), obesity was associated with significant harmful effects on sustaining metabolic health (OR = 0.14, 95% CI [0.03–0.53], $p = 0.004$). In the second sensitivity analysis (Table 6), 128 participants with completed follow-ups were defined as MHOO. When the cut-point of obesity was lowered to 25 $kg/m^2$, the detrimental effect of obesity did not reach the level of significance ($\alpha = 0.007$). However, increased waist circumference remained a risk factor among participants with MHOO for maintenance of metabolic health (OR = 0.92, 95% CI [0.89–0.95], $p < 0.001$).

**Table 4** Factors related to MHOO individuals maintaining metabolic health, stratified by baseline age ($n = 89$).

| | Baseline age <48 years ($n = 41$) | | Baseline age ≥48 years ($n = 48$) | |
|---|---|---|---|---|
| | OR (95% CI) | P | OR (95% CI) | P |
| Baseline age | 0.74 (0.57, 0.95) | 0.02 | 0.71 (0.55, 0.92) | 0.009 |
| Women | 1.00 (0.19, 5.22) | 1.00 | 0.35 (0.10, 1.17) | 0.09 |
| Education | | | | |
| Illiterate/elementary/senior/junior high school | Ref. | | Ref. | |
| University and above | 3.37 (0.69, 16.3) | 0.13 | 0.48 (0.17, 1.37) | 0.17 |
| Cigarette smoking[a] | | | | |
| Non-smoker | Ref. | | Ref. | |
| Smoker | 0.11 (0.01, 0.89) | 0.04 | 0.42 (0.08, 2.13) | 0.29 |
| Ex-smoker | 0.54 (0.03, 8.31) | 0.66 | 0.04 (0.003, 0.59) | 0.02 |
| Alcohol consumption[a] | 0.62 (0.14, 2.83) | 0.54 | 1.78 (0.41, 7.64) | 0.44 |
| Physical activity[a,b] | | | | |
| Low | Ref. | | Ref. | |
| Moderate | 2.15 (0.50, 9.21) | 0.30 | 0.44 (0.14, 1.41) | 0.17 |
| High | 1.19 (0.13, 10.66) | 0.88 | 4.00 (0.94, 16.99) | 0.06 |
| BMI[a] (kg/m$^2$) | | | | |
| BMI <24 | Ref. | | Ref. | |
| 24 ≤ BMI <27 | 2.42 (0.21, 27.77) | 0.48 | 0.36 (0.06, 2.37) | 0.29 |
| BMI ≥ 27 | 0.04 (0.002, 0.82) | 0.04 | 0.08 (0.01, 0.63) | 0.02 |
| Follow-up year | 0.04 (0.005, 0.32) | 0.003 | 0.44 (0.16, 1.25) | 0.13 |

**Notes.**

Abbreviations: MHOO, metabolically healthy overweight/obesity; OR, odds ratio; CI, confidence interval; BMI, body mass index.

[a]Cigarette smoking, alcohol consumption, physical activity, and groupings of BMI were measured at baseline and follow-up.

[b]Physical activity was evaluated by the International Physical Activity Questionnaire (IPAQ) Short-Form, Taiwan version.

The analyses were performed using generalized estimating equations (GEE) models.

# DISCUSSION

This study found harmful factors on maintenance of metabolic health for individuals with MHOO to be: older age, increased years until follow-up, a BMI of $\geq 27$ kg/m$^2$, and an increased waist circumference. This study provided Asian data for the factors associated with maintenance of metabolic health for individuals with MHOO.

The cut-points for overweight and obesity were ethnicity-specific. In its main analyses this study used the cut-points recommended by the Taiwanese official public health agency, and in its sensitivity analyses it used the World Health Organization's recommendation for the Asian population. Both analyses indicated that an increased waist circumference is harmful to the maintenance of metabolic health for individuals with MHOO. Obesity was also a risk factor, though it was significant only under the higher cut-point of 27 kg/m$^2$. In part, our results were similar to previous studies. *Hwang et al. (2015)* followed 85 Japanese Americans with MHO for 10 years. In Hwang's study, obesity was defined as a BMI $\geq 25$ kg/m$^2$. They found male, high HDL-C, lower level of fasting plasma insulin, and lower visceral abdominal fat were factors associated with sustained metabolic health. *Achilike et al. (2015)* followed 275 Mexican Americans and non-Hispanic whites with MHO and

**Table 5   Sensitivity analysis 1, using a different definition of metabolic health[a] to assess the factors related to MHOO individuals maintaining metabolic health ($n = 89$).**

|  | OR (95% CI) | P |
| --- | --- | --- |
| Baseline age | 0.93 (0.86, 0.99) | 0.03 |
| Women | 0.53 (0.24, 1.14) | 0.10 |
| Education |  |  |
| Illiterate/elementary/senior/junior high school | Ref. |  |
| University and above | 0.92 (0.45, 1.88) | 0.81 |
| Cigarette smoking[b] |  |  |
| Non-smoker | Ref. |  |
| Smoker | 0.48 (0.16, 1.46) | 0.20 |
| Ex-smoker | 0.43 (0.13, 1.44) | 0.17 |
| Alcohol consumption[b] | 1.00 (0.41, 2.44) | 0.99 |
| Physical activity[b,c] |  |  |
| Low | Ref. |  |
| Moderate | 0.84 (0.40, 1.74) | 0.64 |
| High | 2.63 (0.94, 7.39) | 0.07 |
| BMI[b] (kg/m$^2$) |  |  |
| BMI <24 | Ref. |  |
| $24 \leq$ BMI <27 | 1.05 (0.28, 3.90) | 0.94 |
| BMI $\geq$ 27 | 0.14 (0.03, 0.53) | 0.004 |
| Follow-up year | 0.50 (0.25, 1.01) | 0.05 |

**Notes.**

Abbreviations: MHOO, metabolically healthy overweight/obesity; OR, odds ratio; CI, confidence interval; BMI, body mass index.

[a] Metabolic health was defined as (1) the absence of cardiometabolic diseases and (2) the presence of $\leq 2$ metabolic risk factors, including hypertension, hyperglycemia, hypertriglyceridemia, low serum high-density lipoprotein cholesterol, and abdominal obesity.

[b] Cigarette smoking, alcohol consumption, physical activity, and groupings of BMI were measured at baseline and follow-up.

[c] Physical activity was evaluated by the International Physical Activity Questionnaire (IPAQ) Short-Form, Taiwan version. The analysis was performed using generalized estimating equations (GEE) models.

had a median follow-up time of 7.8 years. Their results indicated that greater increases in BMI, waist circumference, fasting glucose, fasting insulin, and triglycerides were associated with progression to metabolic abnormalities. *Appleton et al. (2013)* analyzed 188 MHO people and had a median follow-up time of 8.2 years. They reported younger age, smaller waist circumference, and higher socioeconomic status were associated with maintenance of metabolic health. Another cohort study analyzed the UK Clinical Practice Research Datalink and had a mean follow-up time of 9.4 years; it reported female, younger age, lower BMI category, and non-smoker to be predictors of sustained metabolic health for MHO individuals (*Moussa et al., 2019*). Integrating existing literature and our results, obesity and a wider waist circumference undoubtedly are detrimental factors on the maintenance of metabolic health among MHO people. Other factors, such as sex, socioeconomic status, smoking, alcohol consumption, and physical activity seem to lack consistent associations with sustained metabolic health.

In our study 42.7% of individuals with MHOO could not maintain a metabolically healthy status by the time of their 9-year follow-up. According to the report from

**Table 6  Sensitivity analysis 2, using the Asian population-specific cut-points[a] for overweight and obesity to assess the factors related to MHOO individuals maintaining metabolic health ($n = 128$).**

| | Model 1 | | Model 2 | |
| --- | --- | --- | --- | --- |
| | OR (95% CI) | *P* | OR (95% CI) | *P* |
| Baseline age | 0.97 (0.92, 1.02) | 0.22 | 0.96 (0.91, 1.01) | 0.11 |
| Women | 0.85 (0.44, 1.64) | 0.62 | 0.74 (0.38, 1.46) | 0.39 |
| Education | | | | |
| Illiterate/elementary/ senior/junior high school | Ref. | | Ref. | |
| University and above | 1.31 (0.72, 2.39) | 0.37 | 1.47 (0.81, 2.66) | 0.20 |
| Cigarette smoking[b] | | | | |
| Non-smoker | Ref. | | Ref. | |
| Smoker | 0.64 (0.26, 1.54) | 0.32 | 0.68 (0.28, 1.63) | 0.38 |
| Ex-smoker | 0.52 (0.18, 1.49) | 0.23 | 0.60 (0.21, 1.74) | 0.35 |
| Alcohol consumption[b] | 1.32 (0.63, 2.79) | 0.47 | 1.34 (0.62, 2.89) | 0.46 |
| Physical activity[b,c] | | | | |
| Low | Ref. | | Ref. | |
| Moderate | 1.06 (0.58, 1.94) | 0.84 | 1.08 (0.58, 1.99) | 0.81 |
| High | 1.64 (0.73, 3.70) | 0.23 | 1.50 (0.67, 3.37) | 0.33 |
| Follow-up year | 0.69 (0.41, 1.18) | 0.17 | 0.72 (0.42, 1.23) | 0.23 |
| BMI[b] (kg/m$^2$) | | | | |
| BMI <23 | Ref. | | – | |
| 23 ≤ BMI <25 | 0.66 (0.19, 2.24) | 0.50 | – | |
| BMI ≥ 25 | 0.21 (0.06, 0.73) | 0.01 | – | |
| Waist circumference[b] (cm) | – | | 0.92 (0.89, 0.95) | <0.001 |

**Notes.**

Abbreviations: MHOO, metabolically healthy overweight/obesity; OR, odds ratio; CI, confidence interval; BMI, body mass index.

[a]In accordance with the Asian population-specific cut-points recommended by the World Health Organization, overweight was defined as 23 kg/m$^2$ ≤ BMI <25 kg/m$^2$ and obesity was defined as BMI ≥ 25 kg/m$^2$.

[b]Cigarette smoking, alcohol consumption, physical activity, groupings of BMI, and waist circumference were measured at baseline and follow-up.

[c]Physical activity was evaluated by the International Physical Activity Questionnaire (IPAQ) Short-Form, Taiwan version.

The analyses were performed using generalized estimating equations (GEE) models.

*Appleton et al. (2013)* nearly 32.0% of their 847 European adults developed two or more cardiometabolic risk factors after a median follow-up time of 8.2 years. A cohort study from China followed 5,850 adults with MHOO for a median follow-up time of 10.0 years, and in that study 2,455 (42.0%) individuals became metabolically unhealthy (*Gao et al., 2020*). *Mongraw-Chaffin et al. (2018)* analyzed data from the United Status and reported 48% of their 1,051 individuals with metabolically healthy obesity at baseline developed metabolic syndrome during the follow-up period of 12.2 years. The transition rate for overweight people and obese people from healthy to unhealthy metabolic status increased over time. This result corresponds with our finding that an increased number of years until follow-up was negatively associated with the maintenance of metabolic health for MHOO individuals.

For overweight people and obese people, maintaining metabolic health helps to avoid adverse health outcomes. Studies investigating populations from the United States, China, and countries in Europe have reported the risks of cardiovascular events, incident diabetes, and all-cause mortality do not increase in overweight/obese subjects with sustained metabolic health compared with those who remain metabolically healthy normal-weight over time (*Appleton et al., 2013*; *Mongraw-Chaffin et al., 2018*; *Gao et al., 2020*). However, differing results have been published. Research from the Nurses' Health Study followed 21,332 women with MHOO for 20 years (*Eckel et al., 2018*). Even with maintenance of metabolic health, the risk of total cardiovascular disease was higher in metabolically healthy obese women as compared with women who were metabolically healthy normal weight throughout the duration of the study (*Eckel et al., 2018*). Despite these particular contrasting results, each of these studies reported MHO individuals had a lower risk of cardiovascular disease than metabolically unhealthy obese people, which reflects the importance of metabolic health for obese people. In our study avoidance of obesity was beneficial for maintenance of metabolically healthy status for MHOO individuals. This result aligns with a randomized control trial that reported exercise- or diet-induced weight loss in MHO subjects improved their cardio-metabolic risk factors and insulin sensitivity (*Janiszewski & Ross, 2010*). Therefore, weight control is always important for overweight people and obese people, even when no metabolic disease is present.

In addition to weight control, a smaller waist circumference was a positive factor in our study for sustained metabolic health in MHOO subjects. The result is consistent with the study from *Appleton et al. (2013)* which reported lower waist circumference is associated with the maintenance of metabolic health in MHO individuals. A study from the English Longitudinal Study of Ageing also reported a similar result. That study followed 409 unhealthy obese adults for 8 years and reported larger increases in waist circumference as a risk factor for remaining metabolically unhealthy (*Hamer et al., 2015*). Abdominal obesity, which represents visceral fat deposits, plays a central role in metabolic disorders. Visceral obesity is related to a pro-inflammatory status and oxidative stress, which result in endothelial dysfunction and subsequent cardio-metabolic diseases (*Ritchie & Connell, 2007*). Thus, in order to maintain metabolic health, overweight people and obese people should dedicate themselves to avoiding abdominal obesity.

For metabolically unhealthy adults, intensive lifestyle interventions, including diet control and exercise, effectively reduce waist circumference and improve metabolic risk factors (*Salas-Salvadó et al., 2019*). One meta-analysis analyzed seven studies that performed lifestyle interventions (length from 2 to 9 months) on obese people and reported that lifestyle interventions resulted in greater benefits for people with metabolically unhealthy obesity, including improvements in fasting glucose, blood pressure, and bodyweight (*Lin et al., 2017*). For MHO individuals, diet and exercise intervention had a significant change in weight reduction but not metabolic parameters (*Lin et al., 2017*). This result may be because the MHO individuals were metabolically healthy, therefore their change in metabolic parameters was unremarkable. However, our study supported that a smaller waist circumference and avoidance of obesity had protective effects on the maintenance of metabolic health for MHOO individuals. Accordingly,

lifestyle interventions designed to result in weight reduction for MHOO people should be beneficial for maintenance of metabolic health. Future studies with a longer intervention period and a larger sample size are needed to investigate the relationship between lifestyle modifications and maintenance of metabolic health for MHOO individuals.

In the subgroup analyses, individuals younger than 48 years old with MHOO were more vulnerable to the harmful effects of obesity on maintenance of metabolic health. A prior meta-analysis of 239 studies conducted on four continents observed the risk for all-cause mortality for obesity was higher at a younger age (35–49 years) than an older age (50–69 years) (*Global et al., 2016*), implying the impact of BMI on health is modified by age. However, it may be because of a limited sample size that the interactive effects of age and obesity did not reach a significant level in our analysis. Future research with a larger cohort is needed to examine the effect modification of age and obesity on maintenance of metabolic health.

This study had some limitations. First, our study population was northern Taiwanese people. Even in Taiwan, the prevalence of obesity, diet pattern, and health behaviors differed among various regions in Taiwan (*Yeh, Chang & Pan, 2011*). The generalizability of our results may be limited to other populations. Second, during the follow-up period, nearly half of the participants with MHOO at baseline were lost to follow-up and, as a result, a selection bias may exist. However, except that study participants had a higher mean blood pressure, the demographic characteristics between participants who completed follow-up and those lost to follow-up were similar. Maintenance of metabolic health is more likely to be difficult in the study population, so the related factors are more reliable and pertinent. Third, the information on social-economic status was incomplete. We used marital status and level of education to represent social factors. However, marital status was not adjusted in the final models because nearly all participants were married. Fourth, insulin sensitivity was not included in the criteria for metabolic health data because insulin sensitivity data was unavailable. We conducted a sensitivity analysis with a different definition of metabolic health to examine our results, and this analysis yielded similar results. Therefore, our results are comparable to other studies. Fifth, the number of participants with MHOO was small and the data on diet quality was lacking. Further research with a larger representative sample and adjusted for diet quality is warranted in the future.

Despite these limitations, this study had several strengths. First, the participants were a randomized sample from actual communities, which reflects the representativeness of the study population. Second, the longitudinal design of this study allowed for confidence in making causal inferences based on the evidence. Third, this study provided Taiwanese data for the factors associated with maintenance of metabolic health for overweight people and obese people. By contributing native Asian data for this pressing health issue, we have broaden the horizon of research about MHOO. Also, we used Asian population-specific cut-points for overweight and obesity in the sensitivity analysis, which facilitates comparisons with other Asian populations.

## CONCLUSIONS

This study observed obesity and an increased waist circumference to be detrimental factors in the maintenance of metabolic health among Taiwanese adults with MHOO. For overweight people and obese people, control of weight and waist circumference should remain a priority even when there are no metabolic disorders present. Future research is warranted to examine the effects of lifestyle interventions on the maintenance of metabolic health for MHOO individuals.

## ACKNOWLEDGEMENTS

The authors would like to thank all participants in this study.

### Funding

This work was supported by Taipei Veterans General Hospital, Taipei, Taiwan (grant number: V96S3–001, −002, −005; V97S3 –001, −002, −005; V98S3 –001, −002, −005; V99S3 –001, −002, −005; V106E-005-1; V106E-005-2). The funders had no role in study design, data collection and analysis, decision to publish, or preparation of the manuscript.

### Grant Disclosures

The following grant information was disclosed by the authors:
Taipei Veterans General Hospital, Taipei, Taiwan: V96S3–001, V96S3–002, V96S3–005, V97S3 –001, V97S3 –002, V97S3 –005, V98S3 –001, V98S3 –002, V98S3 –005, V99S3 –001, V99S3 –002, V99S3 –005, V106E-005-1, V106E-005-2.

### Competing Interests

The authors declare there are no competing interests.

### Author Contributions

- Yi-Hsuan Lin conceived and designed the experiments, analyzed the data, prepared figures and/or tables, and approved the final draft.
- Hsiao-Ting Chang conceived and designed the experiments, prepared figures and/or tables, and approved the final draft.
- Yen-Han Tseng analyzed the data, authored or reviewed drafts of the paper, and approved the final draft.
- Harn-Shen Chen performed the experiments, prepared figures and/or tables, and approved the final draft.
- Shu-Chiung Chiang performed the experiments, prepared figures and/or tables, authored or reviewed drafts of the paper, and approved the final draft.
- Tzeng-Ji Chen conceived and designed the experiments, authored or reviewed drafts of the paper, and approved the final draft.
- Shinn-Jang Hwang performed the experiments, authored or reviewed drafts of the paper, and approved the final draft.

## Human Ethics

The following information was supplied relating to ethical approvals (i.e., approving body and any reference numbers):

This study was approved by the Research Ethics Committee of Taipei Veterans General Hospital, Taipei, Taiwan on March 17, 2017 (Protocol Code: 2017-01-009BCF, 97-12-06A).

## Data Availability

The raw data is available in the Supplementary Files.

## Supplemental Information

Supplemental information for this article can be found online at http://dx.doi.org/10.7717/peerj.13242#supplemental-information.

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
