# Peer review of "Factors related to overweight and obese populations maintaining metabolic health"

_PeerJ, doi:10.7717/peerj.13242_

## Round 0.1 · original submission · Major Revisions

Please address all the reviewers' concerns, especially in relation to the study design and the important drop off in the study, evidenced by both reviewers, which may determine a selection bias.

Reviewer 1 ·

Basic reporting

Thanks for giving me the opportunity for reviewing this interesting manuscript. It discovers important features for maintaining metabolic health for people with MHOO. I believe the manuscript itself is clear and unambiguous with relevant literature provided.

Experimental design

This prospective cohort study is well designed with defined research question. However I have some questions as follows:
1. In Table 4, analysis was performed by stratification of age (48), however I can not tell the importance of this stratification from the manuscript, nor there is discussion of this result. Please explain why you use this cut-point and its clinical significance.
2. In Line 295, the author mentioned "...the participants were randomized sampled from....." However I can't understand the randomization process from the study design section.
3. There is no inclusion specific inclusion criteria or exclusion criteria mentioned in the manuscript, please provide inclusion and exclusion criteria in the study design section with more detail.
4. I think a flow chart regarding the enrollment of subjects, the inclusion and exclusion criteria and further follow-up measurements would help readers to better understand this research.

Validity of the findings

This manuscript includes robust data with statistically soundness. The conclusions are well stated with linkage to original research question. However I have a question regarding subjects who dropped out from the study. Originally, 195 participants with MHOO were included at baseline, but only 89 participants completed the follow-up. In other words, more than 50% of participants dropped out and this could cause a selection bias. Please explain the reason why these participants could not finish the follow-up, or you may add this to the limitation section.

Additional comments

Please copyedit your manuscript text to correct grammatical and typographical errors.

Reviewer 2 ·

Basic reporting

1. The main issue here is that the reference for the consensus definition for metabolic syndrome appears incorrect. There is consensus around the definition of metabolic syndrome, resulting in the international harmonized definition of metabolic syndrome. The authors should use that as their primary definition in order to ensure comparability with other studies and other populations. It is understandable to want to assess a more stringent definition of metabolically healthy (especially in populations like this where more participants fall into that category), but this should be done in the context of modifying the harmonized definition. For example, specifying no MetS criteria or only 1 MetS criteria to be “healthy”.
Alberti KG, Eckel RH, Grundy SM, et al. Harmonizing the metabolic syndrome: a joint interim statement of the International Diabetes Federation Task Force on Epidemiology and Prevention; National Heart, Lung, and Blood Institute; American Heart Association; World Heart Federation; International Atherosclerosis Society; and International Association for the Study of Obesity. Circulation. 2009;120(16):1640-1645. doi:10.1161/CIRCULATIONAHA.109.192644

2. Additional care in the language and framing of the results when discussing the direction of the estimates with respect to preserving metabolic health would help the reader keep track of the appropriate interpretations.

3. More data on this topic is needed from different Asian populations living in Asia. This unique population is a major strength of this manuscript. I commend the authors for their appropriate use of population specific cut-points for overweight and obesity. The discussion could be strengthened by putting comparison of these results with that from prior work in this context. As mentioned above, the large proportion of this population that meets a more stringent defintion of metabolically healthy obesity is worth mentioning as many western cohorts have too few particiapnts in this category to investigate.

Experimental design

The study suffers from substantial threats to validity and clarity of methods could be improved. The suggestions are in order of importance.

1. The issue of selection bias from massive loss to follow up cannot be ignored. At the very least a table comparing baseline values of those who remain in the study compared to those who dropped out is needed with commentary on the direction and suspected magnitude of the bias. If possible, assessment of competing risks including information on how many participants did not complete the follow up because they died would be useful. While the original sample being random and fully generalizable is wonderful, if the loss to follow up is differential, then this study will be impacted.

2. Additional specifics about the analysis are needed for clarification. Were data from the follow-up included as exposures in the models? This is not clear and changes the interpretation of the results. For instance, why is the reference group for BMI a category that was excluded at baseline? Additionally, that category is only represented by 15 participants during follow-up, making this a statistically weak reference group in addition to being inconsistent with the original eligibility criteria. Including changes over time, but not showing whether the results for change over time differ from baseline levels is confusing.

3. Stepwise variable selection is not an appropriate method of variable selection for causal inference. An epidemiologic approach that considers the potential causal relationships of the variables including confounders, mediators, and colliders is better. For instance, removing socioeconomic factors from the model because they are not significant, does nothing to address their potential as confounders of the other variables.

4. Please define “Follow-up year”. Does this imply longer duration of follow-up for every participant? More information on the range of follow-up would be informative here.

5. Were those who lost weight more likely to be older? Could this be a source of reverse causality with those who are sicker or frailer more likely to close weight? Similarly, including the number of participants with pre-existing comorbidities in Table 1 would be helpful. Was prevalent CVD excluded or adjusted for? The inclusion of participants with prevalent CVD could impact the results as the relationship between risk factors and outcomes often differs for incident and prevalent cases.

Validity of the findings

The conclusions are well linked to the results; however, the validity of the results may be threatened by the biases described in the Design section. Once those threats to validity are addressed, the results may be well supported.

---

## Round 0.2 · Minor Revisions

Please address the remaining minor issues raised by reviewer 1.

Reviewer 1 ·

Basic reporting

I believe the revised manuscript itself is clear and unambiguous with relevant literature provided.

Experimental design

Thanks for the authors to answer my questions. However I still have a doubt concerning Table 2, which compared health behaviors and cardiometabolic profiles of MHOO participants at baseline and follow-up.

In Methods section of your revised manuscript, line 308, you wrote: "The definition of metabolic health......compromised of [1] an absence of cardiometabolic diseases, including hypertension, hyperlipidemia, Type 2 diabetes, coronary artery diseases, cerebral vascular diseases, and peripheral vascular diseases and [2] the presence of ≤ 1 of the following cardiometabolic profiles......" Those who did not satisfy the definition of MHOO in this study were excluded.

However, in Table 2, there were still 26 of 89 participants who had Hypertension, 7 with Hyperglycemia, 7 with Low HDL and 10 with elevated TG. These results are quite confused with exclusion criteria. Please explain.

Validity of the findings

Thanks for the authors to answer my questions. In discussion section, line 1154, you wrote: "our study population was Taiwanese people, so the generalizability of our results may be limited to other populations." That is true, however I still want to remind that the obesity prevalence, diet pattern and health behaviors were very different between different regions across Taiwan. As followed reference:

Yeh CJ, Chang HY, Pan WH. Time trend of obesity, the metabolic syndrome and related dietary pattern in Taiwan: from NAHSIT 1993-1996 to NAHSIT 2005-2008. Asia Pac J Clin Nutr. 2011;20(2):292-300.

Thus, since your study enrolled only participants living in the Northern region of Taiwan, there is a chance that different results would be discovered if further study with participants from other regions of Taiwan is made.

Reviewer 2 ·

Basic reporting

No Comment

Experimental design

No comment

Validity of the findings

My only final comment is that adjustment for multiple testing is not necessary in this manuscript and can be removed. Adjustment for multiple testing assumes that each test is independent, and that is not the case here.

---

## Round 0.3 · accepted · Accept

The authors have satisfactorily addressed the remaining issues raised by the reviewers.